# Various Flexible Fixation Techniques Using Suture Button for Ligamentous Lisfranc Injuries: A Review of Surgical Options

**DOI:** 10.3390/medicina59061134

**Published:** 2023-06-12

**Authors:** Young Yi, Sagar Chaudhari

**Affiliations:** 1Department of Orthopedic Surgery, Seoul Foot and Ankle Center, Inje University Seoul Paik Hospital, 85, 2-ga, Jeo-dong, Jung-gu, Seoul 04551, Republic of Korea; 2Department of Orthopedic Surgery, K. B. Bhabha Hospital, Bandra, Mumbai 400050, Maharashtra, India; drsagarchaudhari@yahoo.com

**Keywords:** Lisfranc joint injury, flexible fixation technique, suture button, TightRope

## Abstract

Contrary to Lisfranc joint fracture-dislocation, ligamentous Lisfranc injury can lead to additional instability and arthritis and is difficult to diagnose. Appropriate procedure selection is necessary for a better prognosis. Several surgical methods have recently been introduced. Here, we present three distinct surgical techniques for treating ligamentous Lisfranc employing flexible fixation. First is the “Single Tightrope procedure”, which involves reduction and fixation between the second metatarsal base and the medial cuneiform via making a bone tunnel and inserting Tightrope. Second is the “Dual Tightrope Technique”, which is similar to the “Single Tightrope technique”, with additional fixation of an intercuneiform joint using one MiniLok Quick Anchor Plus. Last but not least, the “internal brace approach” uses the SwiveLock anchor, particularly when intercueniform instability is seen. Each approach has its own advantages and disadvantages in terms of surgical complexity and stability. These flexible fixation methods, on the other hand, are more physiologic and have the potential to lessen the difficulties that have been linked to the use of conventional screws in the past.

## 1. Introduction

Only around 0.20% of all orthopedic injuries are Lisfranc joint injuries, which are an exceptionally uncommon trauma [1]. The severity of the injury is typically described as ranging from a little nondisplaced sprain to an apparent fracture dislocation of the midfoot. Ligamentous Lisfranc injury is typically produced by low energy trauma, making it challenging to identify and treat, in contrast to Lisfranc joint fracture-dislocation, which is typically caused by high energy trauma. If not properly and promptly treated, it may result in instability and post-traumatic arthritis in the patient [2]. Unfortunately, the prevalence of ligamentous Lisfranc injuries is rising as sports and professional athletes become more and more prominent (Figure 1). Numerous activities, including football, gymnastics, horseback riding, and jogging, have been linked to ligamentous Lisfranc injuries [3,4]. However, unlike high intensity injuries, indicators of ligamentous Lisfranc damage might not be as visible. The actual frequency of these injuries may thus be overestimated. Therefore, to diagnose ligamentous Lisfranc injuries, a proper physical examination and radiographs are needed.

After low intensity trauma, any patient who complains of midfoot discomfort should be examined, and Lisfranc damage should be ruled out. The injury mechanism is known to entail an axial stress applied to the forefoot being forcedly abducted or the foot being in a plantar flexed posture [5]. Physical findings that are favorable may include midfoot edema, plantar ecchymosis, difficulty bearing weight, a positive piano key stress test, and the dorsal drawer test of the medial column. Diastasis of the intermetatarsal gap between the first and second metatarsals, enlargement of the space between the medial cuneiform and second metatarsal base, the presence of a second metatarsal base fleck, and dorsal subluxation of the metatarsal base on a lateral view are all significant radiographic findings [6]. Patients with a strong clinical suspicion of Lisfranc injury and ambiguous radiographic results may undergo computerized tomography (CT) or magnetic resonance imaging (MRI).

Computerized tomography (CT) is utilized to more precisely assess the congruity of the TMT joint because it can detect an avulsion fracture that would otherwise go undetected on conventional radiographs (Shim et al.) [7]. According to reports, examining and contrasting the affected and uninjured feet can aid in the diagnosis and treatment of Lisfranc ligament injuries. However, if there is no visible diastasis, subluxation, or dislocation, a minor Lisfranc injury could be overlooked on a typical CT scan. In their study, Kennelly et al. [8] contended that a typical CT scan only sometimes finds further injuries and does not offer any more information for individuals who have a positive weight-bearing film. Because it is a static modality of imaging, the traditional CT scan has a limited function in the diagnosis of mild Lisfranc injuries. A recent study found that a bilateral weight-bearing computed tomography is a good diagnostic tool for identifying Lisfranc injuries under physiological stress and provides a comparison with the unaffected side [8,9]. Patients may have a weight-bearing CT under an ankle block or may be sent to an orthopedic out-patient clinic for a few days following the accident if weight-bearing CT is not possible due to discomfort [8,10].

Dorsal, interosseous, and plantar Lisfranc ligaments are the three types of mild ligamentous injury that may be evaluated by magnetic resonance imaging (MRI). The first and most crucial step in treating a ligamentous Lisfranc injury is an accurate assessment of physical findings and radiography.

## 2. Classifications

Building on the work of earlier categorization methods, including those of Quenu, Kuss, and Hardcastle [11,12], Myerson developed the most well-known classification scheme in 1986. This method tries to group injuries based on the alignment of the joints, the location of the involvement, and the direction of instability. It is not predictive but it does shed light on different damage patterns and probable energy loss mechanisms that give rise to such patterns. This classification approach was found to have a good intra- and interobserver reliability [13]. Myerson recently modified his original categorization approach to include moderate, nondisplaced lesions [14]. He categorized these injuries as type D injuries and further separated them into D1 and D2 injuries. D1 injuries do not require surgical stabilization, in contrast to D2 injuries, which necessitate surgery if the medial cuneiform-second metatarsal gap increases by more than 2 mm. D2L injuries were exclusively ligamentous but D2B injuries were bone avulsions.

Nunley and Vertullo [15] developed a categorization system for sports Lisfranc sprains, which is shown in Table 1. This technique attempted to grade sprains using clinical information, weight-bearing radiographs, and bone scintigram findings. The ability to discriminate between injuries with and without longitudinal arch height collapse, as well as very mild nondisplaced injuries, is a benefit of this categorization system. However, due to the widespread use of magnetic resonance imaging (MRI) as an extra modality to detect moderate injuries, this categorization approach is not usually used. There are no categorization systems in use right now that can both direct care and predict results.

Prior categories, however, had certain drawbacks in that choosing a surgical technique was not made any easier. Therefore, a new categorization for modest Lisfranc injury based on anatomical configuration is required. Medial cuneiform (C1)—second metatarsal bone (M2) ligament damage with diastasis is one possible variation of isolated Lisfranc ligament damage. Other variations include medial cuneiform (C1)—intermediate cuneiform (C2) instability, medial cuneiform (C1)—first metatarsal bone (M1) instability, and intermediate cuneiform (C2)—second metatarsal bone (M2) (Table 2 and Figure 2) [16].

According to the aforementioned types, a single tightrope may be preferred for C1-M2 injuries limited to diastasis [17], where damage with diastasis and the isolated Ligamentous Lisfranc injury are present. If C1–C2 instability is added [18] between medial cuneiform and intermediate cuneiform, a dual tightrope or internal brace may be preferred. Additionally, screw or plate fixation and reconstruction might be taken into consideration when there is joint instability. As a result, when considering flexible fixation surgery for ligamentous Lisfranc damage, surgeons must carefully assess the severity of the injury and the level of joint instability using adequate diagnostic skills, and they must then decide on the most suitable surgical strategy.

## 3. Previous Technical Overview

Over time, the operational method for Lisfranc injuries has evolved. For the best results, anatomic reduction should be performed regardless of the surgical technique [19]. There are several surgical methods available to treat Lisfranc injuries. The primary surgical options are K-wire fixation, screw fixation, adjustable suture button fixation, plate fixation, and arthrodesis [20,21]. When K-wires were used alone, high rates of fixation failure were seen [12]. Although the gold standard of therapy is screw fixation, complications include cartilage injury, screw loosening, head fracture, and the requirement for hardware removal continue to be a few of its issues [19,22,23,24] (Figure 3).

The orthopedic foot and ankle society has seen an increase in the use of suture buttons for ligamentous Lisfranc injuries in order to prevent these problems, which have positive clinical outcomes by better recreating normal architecture and providing greater physiological fixation [25,26].

Three flexible attachment methods will be covered in this article: Three different fixation methods are available: (1) Single TightRope (Arthrex, Naples, FL, USA), (2) Dual TightRope (MiniLok QuickAnchor Plus (DePuy, Mitek, Raynham, MA, USA), and (3) InternalBrace (Arthrex, Naples, FL, USA). Each surgical method will be briefly discussed and accompanied by a case study.

## 4. Operative Technique—Single TightRope

A pneumatic tourniquet was applied to the thigh during the procedure while the patient was in the supine position and under general or spinal anesthesia. Under fluoroscopy, the diastasis between the medial cuneiform and second metatarsal base was assessed. Just lateral to the second metatarsal base, a longitudinal dorsal skin incision was made, allowing for the identification and medial retraction of the extensor hallucis brevis. This was performed to safeguard the underlying neurovascular bundle. A medial skin incision was made over the center of medial cuneiform. A bone-reduction clamp was then used to reduce the damaged Lisfranc joint following this incision. The decrease achieved was then verified and documented using fluoroscopy. To properly reduce the Lisfranc joint, it is crucial to hold the clamp in the right vector. Under the supervision of fluoroscopy, a guide wire was then inserted along the Lisfranc ligament’s length, beginning at the mid-coronal plane of the medial cuneiform and ending just distal to the insertion of the tibialis anterior to the base of the second metatarsal bone.

To avoid damaging the Lisfranc joint, a 3.5 mm reamer was utilized to ream along the guide wire from medial to lateral. Then, a passing pin was used to insert TightRope into the bone tunnel. An oblong button was then placed with the proper tension on the periosteum of the medial cuneiform after the leading button had been placed on the lateral cortex of the second metatarsal base (Figure 4).

Patients with poor bone mineral density, however, run the risk of experiencing suture button migration since the TightRope procedure necessitates reaming a bone tunnel (Figure 5). The Lisfranc joint might be lost or reduced as a result of further issues. Consequently, it is important to choose patients carefully, and extensive bone reaming should be avoided.

## 5. Operative Technique—Dual TightRope

One Mini TightRope and one MiniLok QuickAnchor Plus were used for the Lisfranc fixation procedure employing dual TightRope. A medial skin incision was made across the center of the medial cuneiform and a similar longitudinal dorsal skin incision was performed just lateral to the second metatarsal base. Under the direction of fluoroscopy, a bone-reduction clamp was used to reduce the damaged Lisfranc joint. Surgeons must exercise caution to safeguard the neurovascular bundle dorsally and the anterior tibialis medially, which is equivalent to the single TightRope approach. After using the bone clamp to reduce the Lisfranc joint, the guide wire was initially positioned immediately distal to the second and third metatarsal articulations, from the medial cuneiform to the lateral cortex of the second metatarsal base. After reaming, a passing pin was used to introduce the Mini TightRope into the bone tunnel, making sure the lateral button was securely situated at the lateral cortex of the second metatarsal base. After ensuring that the medial button was firmly positioned at the medial cuneiform, the Mini TightRope was then tightened. Beginning proximally and dorsally to the first inserted Mini TightRope, the guide wire for the MiniLok QuickAnchor Plus crossed the intercuneiform joint between the medial and intermediate cuneiform. The middle of the intermediate cuneiform was reamed using a reamer. After using a bone clamp to narrow the gap between the medial and intermediate cuneiforms, a second anchor was subsequently placed. Pulling the connecting fiber wire gently confirms augmentation (Figure 6).

## 6. Operative Technique—InternalBrace

InternalBrace Lisfranc fixation is advantageous, particularly when intercuneiform instability is noted following a Lisfranc joint fixed. Using a Freer elevator, the stability of the intercuneiform joint was assessed, and an unrestricted passage of the Freer between the medial and intermediate cuneiform verified the instability of the joint. After achieving the proper reduction of the Lisfranc joint, the guide wire was placed similarly to the previous two procedures from the base of the second metatarsal to the medial cuneiform. A 4.75 mm SwiveLock Anchor was then fixed to the center of the medial cuneiform while the proper tension was given, after which the button was inserted at the lateral cortex of the second metatarsal base. The remaining FiberTape suture was then moved over to the intermediate cuneiform through the dorsal incision using a pair of mosquito forceps. The remaining FiberTape was then passed through the hole from the dorsum to the plantar side, starting at the dorsum of the intermediate cuneiform and drilling perpendicularly. A 3.5 mm SwiveLock Anchor was secured to the intermediate cuneiform after the proper amount of tension had been verified via fluoroscopy (Figure 7).

## 7. Other Surgical Technique (Ligament Reconstructions)

Flexible fixation of a Lisfranc injury may be achievable with ligamentous repair as a treatment option. Chronic Lisfranc instability, which is characterized as clinical impairments (pain, instability, or deformity in the tarsometatarsal region) due to post-traumatic symptoms that last longer than six weeks, has historically been treated with this technique [27,28,29]. However, Lisfranc joint arthrodesis is more recommended than ligamentous repair if there is an arthritic alteration in the tarsometatarsal joint [17,29,30].

There have been reports of several surgical procedures for ligamentous reconstruction. A third extensor digitorum longus tendon-based anatomical three bone tunnel reconstruction approach was developed by Nery et al. [31]. The Lisfranc joint has three layers: the dorsal, interosseous, and plantar parts. Biologically, the “Y” plantar ligament between the medial cuneiform and the second to third metatarsals and the Lisfranc ligament between the medial cuneiform and the second metatarsal are the two strongest ligaments [32]. The preparation of three bone tunnels is carried out in a similar manner. Between medial, intermediate, and lateral cuneiform, the first tunnel was created. A second tunnel was created between the second metatarsal and medial cuneiform. The third tunnel was then created between the third metatarsal and medial cuneiform.

Following that, Miyamoto et al. [17] reported a technique for ligamentous restoration using two bundles of the gracilis tendon. The strongest interosseous Lisfranc ligament can be stabilized by creating a bone tunnel between the medial cuneiform and the second metatarsal. The dorsal ligament and interosseous (Lisfranc) ligament, which has been reported as the strongest of these ligaments, were reconstructed, but the plantar ligament between the medial cuneiform and the second metatarsal bone could not be reconstructed due to technical difficulties. As a result, the technique does not achieve true anatomical reconstruction.

The gracilis tendon was used in a four-bundle repair procedure that was presented by De Los Santos-Real et al. [28]. The intermetatarsal joint initially consists of a bone tunnel between the medial cuneiform and second metatarsal, but additional bundles are subsequently produced on the dorsal and plantar portions of the joint. This method is advantageous in that it can recreate the dorsal, interosseous, and plantar layers of the Lisfranc joint to its original shape (Figure 8) [17,28,31]; nevertheless, it has drawbacks such as morbidity at the donor graft location and the need for precise technical skill to remove and deliver sufficient length graft to replicate the three ligament complex bundles.

Depending on the surgeon’s experience, these Lisfranc ligamentous restoration approaches might be taken into consideration for revision surgery if the flexible fixation technique failed when using the same bone tunnel.

## 8. Postoperative Management

On postoperative day 3, a brief leg cast is put on the operated foot if there is no skin issue. Patients are kept off their feet for four weeks. At the fourth week following surgery, the cast is removed, and the patient is given a boot. At this point, gradual weight bearing and range of motion activities are started under the guidance of a medical professional. Patients are encouraged to wear regular shoes, receive physical therapy at two months, and are permitted to completely bear weight with an arch support. Three months after surgery, weight-bearing workouts and athletic endeavors can be started. To ensure the technique’s best possible results, competitive sports can be resumed six months after surgery.

This technique has a benefit over others that use K-wire, screws, or plates since there is no need to remove hardware, theoretically. As a result, early rehabilitation may be started without worrying about late diastasis from poor ligament integrity following metal removal [33].

## 9. Discussion

The metatarsal joint is made up of the medial column, middle column, and lateral column, which together make up a substantial structural component of the mid-foot. It preserves the lateral stability of the joint by forming a characteristic “arch” structure in the cross section between the metatarsal and tarsal bones. The second metatarsal bone is placed between the medial cuneiform and the lateral bone to produce a mortise and tenon structure that acts as a “wedge stone” for the longitudinal stability of the tarsometatarsal joint [34,35]. A number of various sorts of injuries can affect the tarsometatarsal joint, ranging from low-energy ones with minimal subluxations or instability to high-energy ones with a very unstable mid-foot. Low energy injuries occur most commonly and are usually accompanied by ligamentous Lisfranc injuries as a result of axial, rotational, or twisting injuries, especially during movements such as sprinting, leaping, and twisting the weight-bearing foot [36,37,38,39]. The Lisfranc ligament, a strong tissue that joins the medial column to the middle column, is the most important interosseous ligament. Seventy-three percent of feet have a single bundle, while 27% have a double bundle [40].

The severity of a Lisfranc injury can range from ligamentous damage to tarsometatarsal joint fracture-dislocations. Ligamentous Lisfranc injuries are becoming more common as a result of people’s increased involvement in sports. No matter the damage pattern, a sufficient decrease must be attained for the best results.

The best way to repair a Lisfranc injury is still up for debate [41]. Treatment options for Lisfranc injuries range from K-wire fixation to screw fixation to plate fixation to fixation employing different suture buttons [20,23,25,42]. Fixation has traditionally relied on hard fixation methods such as screw fixation or K-wire fixation. Screw fixation has been demonstrated to offer more biomechanical stability than K-wire fixation [41]. The higher risk of arthritis brought on by articular injury and the probable necessity for hardware removal are drawbacks of transarticular screw fixation [22,23,42]. In actuality, all stiff type fixations restrict mobility in the damaged foot’s medial column, leading to discomfort or screw breakage during demanding exercises. Additionally, if a patient’s fixation duration was very brief, it may have contributed to Lisfranc joint dissociation when the screw was removed [42].

For ligamentous, lower-energy injuries, the authors favor maintaining the midfoot articulations. When using various suture button procedures to treat ligamentous Lisfranc injury, authors have seen good clinical and radiological results. By using biological replacements, a notion of non-rigid fixation in Lisfranc joint damage has been put forth. The same idea is used in Fiberwire devices, which avoid sacrificing the autologous tendons and their associated morbidities [43,44].

Single mini-TightRope Lisfranc fixations have had positive clinical outcomes [25,45]. While reducing the Lisfranc joint’s anatomical and physiological dimensions, mobility is preserved. Comparative to the other two suture button procedures, this one is rather straightforward. However, it is possible to see subsidence and the displacement of the buttons into the brand-new tunnel.

Additionally, research on cadavers demonstrates that a single TightRope does not offer enough stability in comparison to screw fixation [46]. The intercuneiform joint between the medial and intermediate cuneiform benefits from the insertion of a second TightRope because it offers more rotational and compressive stability, which prevents Lisfranc joint diastasis from occurring again [18]. The Lisfranc and intercuneiform joints may suffer articular injury with this dual TightRope method, which is more difficult to execute than a single TightRope approach.

Using a 1.6 mm K-wire instead of a 3.5 mm drill bit, InternalBrace has the benefit of reducing articular injury when compared to the suture button procedure. Additionally, it stabilizes the intercuneiform joint as well as the Lisfranc joint, similar to the dual TightRope approach. The dorsal cortex of the medial and intermediate cuneiform, however, may become irritated by Fiberwire. It should be highlighted that InternalBrace Lisfranc fixation is a relatively new procedure, and further research will be required to evaluate its clinical efficacy and failure rate [47] (Table 3).

Skin issues, infections, loss of reduction, neurovascular damage, and post-traumatic arthritis, which may be a possible source of persistent pain, are possible complications of these procedures after surgical therapy. In order to avoid the “N” wrinkle of the implanted suture and reduce friction between the suture and the bone tunnel during foot movement, which ultimately prevents the disruption of the suture, it is also important to prevent the alteration of the bone tunnel between the medial cuneiform bone and the second metatarsal bone. Additionally, while knotting in the medial surface of the medial cuneiform bone, great care should be given to ensuring that the knot is not too big as this might quickly result in a subcutaneous foreign body response and inflammation. Patients with underlying infections, considerable soft tissue edema, medical conditions, peripheral vascular disease, severe bone comminution, and Charcot neuropathy are also often contraindicated for these treatments.

## 10. Conclusions

For the treatment of ligamentous Lisfranc injuries, various flexible fixation methods can be utilized as an alternative to the screw fixation approach. With early rehabilitation and range-of-motion, fixation utilizing a tightrope and internal brace better preserves the natural anatomy and is therefore more physiological. Additionally, it avoids subsequent surgery for hardware removal and lessens the possibility of articular injury brought on by screw fixation.

## Figures and Tables

**Figure 1 medicina-59-01134-f001:**
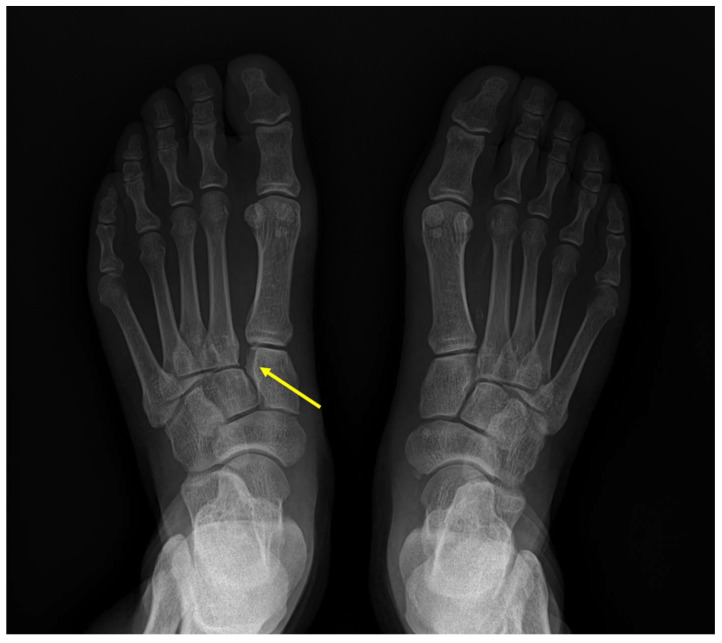
Standing plain radiographic image of bilateral foot. Diastasis between medial cuneiform and 2nd metatarsal base is observed on left foot (yellow arrow).

**Figure 2 medicina-59-01134-f002:**
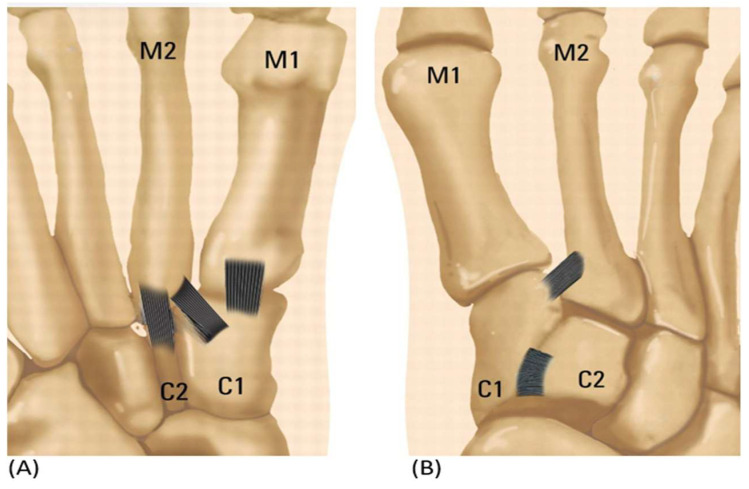
(**A**) A simplified illustration showing the anatomy of plantar sided ligamentous instability. (**B**) Isolated Ligamentous Lisfranc injuries reveal ligamentous instability in the dorsum.

**Figure 3 medicina-59-01134-f003:**
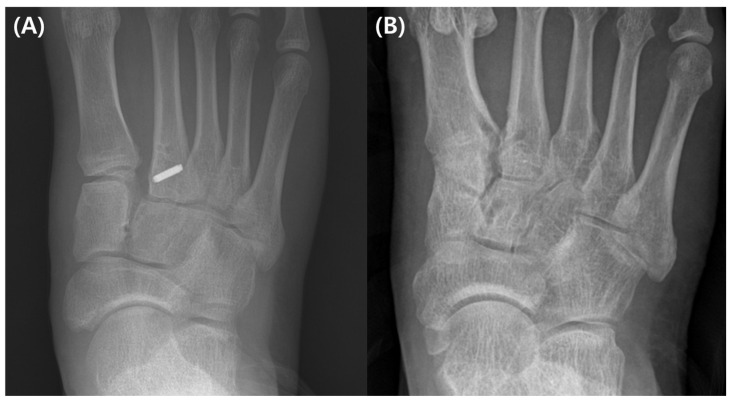
(**A**) A ligamentous Lisfranc injury has been treated with a conventional screw fixation. (**B**) A follow-up fluoroscopic picture taken following implant removal reveals decreased diastasis but arthritic change.

**Figure 4 medicina-59-01134-f004:**
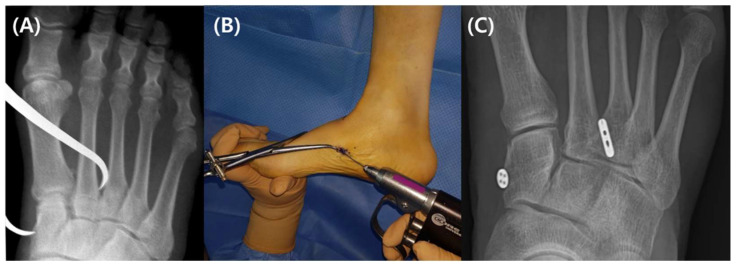
(**A**) A reduction clamp operated under a fluoroscopic image intensifier was used to diminish the diastasis between the medial cuneiform and second metatarsal base. (**B**) The Lisfranc joint was traversed using a guide wire. (**C**) An oblong button was positioned medial to the center of the medial cuneiform and an endobutton was positioned at the lateral cortex of the second metatarsal base.

**Figure 5 medicina-59-01134-f005:**
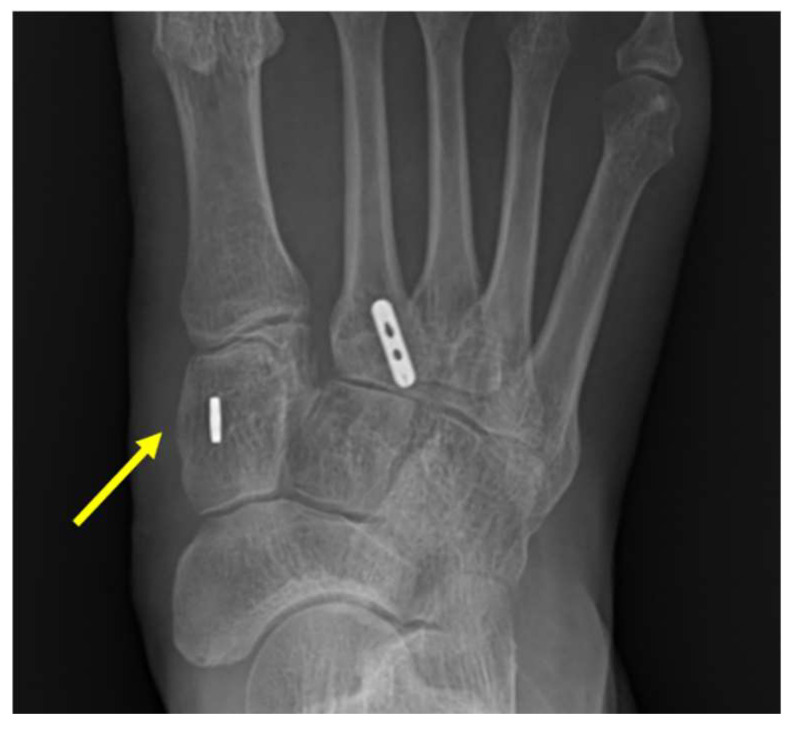
Recurrent diastasis of Lisfranc joint is observed due to internal migration of oblong button (yellow arrow).

**Figure 6 medicina-59-01134-f006:**
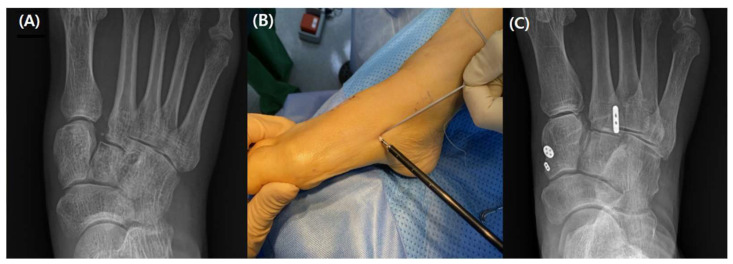
(**A**) A plain radiograph taken prior to surgery demonstrating diastasis of the Lisfranc joint (**B**) Mini TightRope put into the medial cuneiform (**C**) MiniLok QuickAnchor Plus anchor implanted more proximally than Mini TightRope.

**Figure 7 medicina-59-01134-f007:**
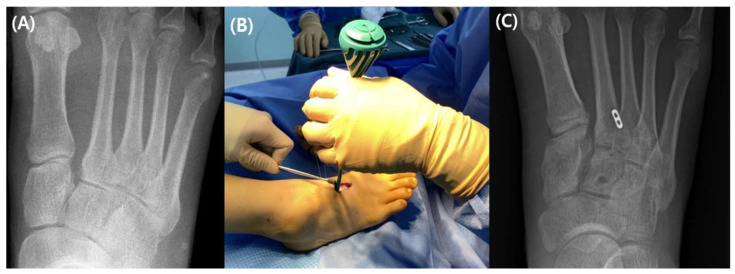
(**A**) A preoperative fluoroscopic image demonstrating the diastasis of the Lisfranc joint and the intercuneiform joint (**B**) A postoperative fluoroscopic image demonstrating the reduced Lisfranc joint and intercuneiform joint (**C**).

**Figure 8 medicina-59-01134-f008:**
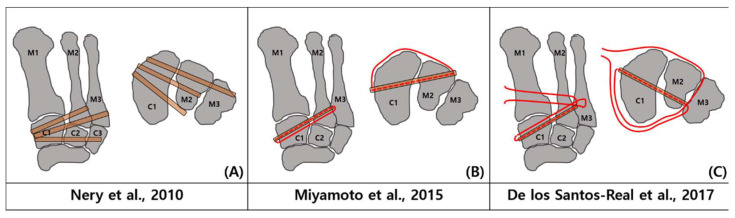
(**A**) third extensor digitorum longus tendon was employed in a triple bone tunnel by Nery et al. [31] to repair the Lisfranc ligament in patients with chronic Lisfranc injury who did not develop arthritis. (**B**) Gracilis tendon was used in a 2-bundled method by Miyamoto et al. [17] (**C**) De los Santos-Real et al. [28] built a 4-bundled repaired ligament using gracilis tendon.

**Table 1 medicina-59-01134-t001:** Nunley and Vertullo classification of athletic Lisfranc injuries [15].

Type	Radiographic Diastasis	Radiographic Loss of Arch Length	Description
Stage I	None	None	- Ligament sprain. - Patients are unable to play sports. - Positive bone scintigram.
Stage II	1–5 mm	None	- Injury of the Lisfranc ligament accompanied with elongation or rupture.
Stage III	>5 mm	Decrease in distance between plantar base of the fifth metatarsal and plantar medial cuneiform.	- Progression of stage II accompanied by damaged plantar tarsometatarsal ligament and joints and potential fracture

**Table 2 medicina-59-01134-t002:** Classification of isolated ligamentous Lisfranc injuries based on anatomical configuration [16].

Type	Description
C1–M2	Medial cuneiform—second metatarsal bone damage with diastasis
C1–C2	Medial cuneiform—intermediate cuneiform instability
C1–M1	Medial cuneiform—first metatarsal bone instability
C2–M2	Intermediate cuneiform—second metatarsal bone instability

**Table 3 medicina-59-01134-t003:** Advantages and disadvantages of three flexible fixation technique.

Surgical Technique	Advantages	Disadvantages
Single TightRope Technique	- Preservation of mobility on Lisfranc joint - Relatively simple procedure [45]	- Relatively weak stability [17]
Dual TightRope Technique	- Higher resistance to rotational force [18]	- Relatively challenging procedure [18]
Internal brace Technique	- Minimal articular damage on Lisfranc joint [48]	- Irritation on dorsal cortex of cuneiforms [49]

## Data Availability

Not applicable.

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
