# Peer review of "Various Flexible Fixation Techniques Using Suture Button for Ligamentous Lisfranc Injuries: A Review of Surgical Options"

_medicina, 2023, doi:10.3390/medicina59061134_

Round 1

Reviewer 1 Report

This is an excellent review work. However, I would like to make a few comments that, I hope, will be helpful to the authors.

-Abstract: it seems that the authors are using new techniques, although they are not. If so, would this be a review article or a clinical trial? I would like this term to be clarified and the wording improved.

- Introduction: an excellent introduction to Lisfranc fracture-dislocation. However, the authors would like to focus more on ligamentous injury. Why don't they focus more on it? Although it is a great exposition of pathology, I think they should focus more on ligamentous injury. 

- Classifications: Totally agree with the authors regarding the advantages and disadvantages of the Meyerson or Nunley and Vertulio classifications. In Ln 92 and following they make a proposal for classification. I think they should do it. I think they should present a Table with this proposal and equate it with Table 1.

- Techniques (3,4,5,6): the description of the different techniques is praiseworthy and with an appropriate iconography. However, I miss the bibliographical references that support the assertions of the authors. They appear exclusively on screw fixation. 

- Technique 7: perfectly documented, both bibliographically and ichnographically. 

- Postoperative management: I would encourage the authors to be more extensive, especially differentiating the times and care according to the technique. 

- Discussion: I would like the authors to guide potential readers more decisively, and not limit themselves to describing the advantages or disadvantages of the techniques. If they have experience, they should give their opinion. Regarding complications (Ln 287 and following), they are the same as any other surgery and I would put more weight on Charcot neuropathy. 

Best regards.

Author Response

Respected Reviewer, 

First of all, I would like to thank you for providing such valuable comments on our article. I have done my best to meet your suggestions and uploading a revised version of the article with this message. kindly go through it at your convenience. 

thanking you 

regards 

Reviewer 2 Report

First of all, I would like to thank the authors and editors for the possibility to review this article. In my opinion problem described in this work is very difficult and doesn't have well documented recommendation how to treat different types of Lisfranc injuries. 

In this paper is well described problem of diagnosis and difficulty with classification different types of injury. Still one of most common method to treat those patients is screw fixation, but nowadays we are increasingly using flexible/relative stable methods of fixation. Foot is very complex in relation to bone and ligaments structure. Stable fixation often lead to arthrodesis which could give even greater pain. 

Description of three different types of flexible fixation is well prepared. One thing I could give you for consideration is to more precisely describe which type of injuries should be treated with specific method. For example single thightrope technique for cuneiform medial/2nd metatarsal instability, dual tightrope technique of intercuneiform instability.

Author Response

Respected Reviewver, 

First of all, I would like to thank you for providing me with your valuable comments. I have done my best to meet your suggestions. Attaching a revised version with necessary corrections.

thanking you 

Regards
